# Pathogenicity of Type I Interferons in *Mycobacterium tuberculosis*

**DOI:** 10.3390/ijms24043919

**Published:** 2023-02-15

**Authors:** Akaash Mundra, Aram Yegiazaryan, Haig Karsian, Dijla Alsaigh, Victor Bonavida, Mitchell Frame, Nicole May, Areg Gargaloyan, Arbi Abnousian, Vishwanath Venketaraman

**Affiliations:** 1College of Osteopathic Medicine of the Pacific, Western University of Health Sciences, Pomona, CA 91766, USA; 2Graduate College of Biomedical Sciences, Western University of Health Sciences, Pomona, CA 91768, USA

**Keywords:** tuberculosis, mycobacteria, interferons, interleukins, tumor necrosis factor

## Abstract

Tuberculosis (TB) is a leading cause of mortality due to infectious disease and rates have increased during the emergence of COVID-19, but many of the factors determining disease severity and progression remain unclear. Type I Interferons (IFNs) have diverse effector functions that regulate innate and adaptive immunity during infection with microorganisms. There is well-documented literature on type I IFNs providing host defense against viruses; however, in this review, we explore the growing body of work that indicates high levels of type I IFNs can have detrimental effects to a host fighting TB infection. We report findings that increased type I IFNs can affect alveolar macrophage and myeloid function, promote pathological neutrophil extracellular trap responses, inhibit production of protective prostaglandin 2, and promote cytosolic cyclic GMP synthase inflammation pathways, and discuss many other relevant findings.

## 1. Introduction

Tuberculosis (TB) is caused by infection with *Mycobacterium tuberculosis (M. tb)* and is among one of the oldest diseases affecting man, with fossil evidence of spinal TB going as far back as 8000 BC [1,2]. While cases of TB have decreased drastically throughout the 20th century, there was a notable surge following the HIV pandemic in the 1980s [3]. Cases of TB declined in large part due to the help of the Bacillus Calmette–Guerin vaccine, first introduced in 1921, with over 3 billion doses administered since [3,4]. Even with all the advances made in the treatment of TB, approximately 100 million people have died in the past 100 years from TB [5,6].

According to the World Health Organization (WHO), it is estimated that 10.6 million people contracted *M. tb* in 2021, an increase of 4.5% from 2020 [7]. It has been documented that infection with SARS and other viral infections can cause a reactivation of TB in the affected patient [8]. As COVID-19 evolved into a pandemic, the number of undiagnosed cases of TB increased, the number of deaths from TB increased, and the trend of a decline in the number of TB cases per year was halted [9]. Furthermore, as healthcare systems around the world were overloaded due to the effects of the COVID-19 pandemic, TB healthcare services were subject to a decline [10]. By some estimates, an additional 400,000 individuals died from TB because of the added strain from COVID-19 on healthcare systems worldwide [11].

## 2. Background

### 2.1. M. tb: Transmission and Clinical Manifestations

*Mycobacterium tuberculosis* is transmitted to humans via respiratory droplets. The pathogen commonly infects the lungs but is also able to manifest in various other tissues [12]. A cascade of events has been proposed in the propagation and transmission of *M. tb* in humans. In the United States, there is an estimated 4.7% of the population infected with *M. tb* as of 2021 [13]. Per the CDC, as of 2018, 79.6% of cases in the U.S. had pulmonary involvement while only 20.4% were extrapulmonary in nature. It is believed that patients with increasing levels of smear-positive pulmonary TB are the ones at the highest risk of infecting other individuals. Patient populations considered immunocompromised, such as HIV-infected individuals, have increasing rates of morbidity and mortality and, as such, there is a significantly increased *M. tb* incidence in parts of the world where HIV is prevalent [14]. While our knowledge and understanding of *M. tb* transmission have vastly improved in recent years, there is still much to be discovered in the understanding of ways to better limit or stop transmission of this disease.

As stated previously, *Mycobacterium tuberculosis* infection is notoriously known for causing pulmonary symptoms; however, as our knowledge and understanding of the disease has progressed, we now know this is not solely a pulmonary disease. While many cases of *M. tb* infection tend to be asymptomatic, symptomatic patients can experience a wide range of constitutional symptoms including fever (most common), weight loss, night sweats, and persistent cough with or without purulent/blood-stained sputum [15,16]. Another subset of symptoms in *M. tb* infection is described as extrapulmonary TB (EPTB) and includes but is not limited to pleuritic chest pain or lymphadenopathy [16]. In these cases, the spread is usually local and without pulmonary involvement. Other common sites of EPTB include cutaneous tissue, the gastrointestinal system, and bones [17,18]. There are also additional rarer manifestations of *M. tb* infections such as the dissemination of the bacteria to the brain resulting in encephalopathy or encephalitis. This can present with signs of meningitis, white matter edema, and demyelination [19]. Miliary TB is a very rare complication resulting from the disease’s lymphohematogenous dissemination, causing both pulmonary and EPTB manifestations. These patients often experience rapidly changing manifestations due to the massive spread of the bacteria throughout the body [20]. As described, *M. tb* infections manifest themselves in many ways. It is important to consider this as a clinician when identifying infected individuals and formulating an appropriate treatment plan.

### 2.2. M. tb: Pathogenesis of Latent Infection

The pathogenesis of latent TB infection (LTBI) is a complex process involving many different cells, cytokines, as well as the host’s ability to respond to infection. LTBI begins when mycobacteria are ingested via phagocytosis by antigen-presenting cells including macrophages and dendritic cells found along the respiratory tract [21]. The antigen-presenting cells recognize the *M. tb* pathogen-associated molecular patterns (PAMPs), resulting in the initiation of a cascade of cytokines [22]. Alveolar macrophages produce inflammatory cytokines including Tumor Necrosis Factor-alpha (TNF-ɑ), interferon (IFN)-ɣ, interleukin (IL)-1, and IL-6, among others [23]. This release of cytokines and chemokines results in the recruitment of monocytes, neutrophils, and other lymphocytes, all of which create the collective immune response [21]. Dendritic cells engulf tubercle bacilli and migrate towards regional lymph nodes where they present *M. tb* antigens to both immature CD4+ and CD8+ T cells. This “matures” the T cells, allowing them to recognize and combat *M. tb* infection. Once mature, cytokines and chemokines guide T cells back to the site of infection where they join with other immune cells to form a granuloma, resulting in LTBI [21]. Granulomas are a protective reflex of the body’s immune system that contain and limit the replication and spread of *M. tb* infection [22]. It is within granulomas that CD4+ T cells control growth of *M. tb* independently of IFN-γ via nitric oxide-dependent mechanisms. Simultaneously, CD4+ T cells release IFN-γ, augmenting the immune response to infection, ultimately ceasing the progression of the disease. It is not until a local immunosuppressive event takes place that the dormant *M. tb* antigens are able to “break free” of the granuloma formation and cause active infection [21].

### 2.3. M. tb: Pathogenesis of Active Infection

Although very similar to that of LTBI, the pathogenesis of active *M. tb* infection contains key differences. The start of active *M. tb* infection occurs when tubercle bacilli are recognized, just like in LTBI, by macrophages and dendritic cells via their PAMPs and are phagocytosed [21]. While some ingested bacteria are destroyed via the innate immune response, others are able to avoid destruction [24]. Survival of infected macrophages allows *M. tb* infection to propagate and spread throughout the body before activation of the adaptive immune response [25]. Knowledge of the adaptive immune response seen in *M. tb* infection is crucial to understanding its pathogenesis. As previously explained, dendritic cells transport *M. tb* antigens to regional lymph nodes, facilitating CD4+ and CD8+ T cell responses [21]. T cells, along with other inflammatory cytokines and chemokines, are responsible for containing the replication and spread of infection. Aside from immunosuppression, *M. tb* utilizes several other mechanisms that collectively aid in evasion of the host immune responses. Some examples include inhibiting antigen processing by major histocompatibility complex class II (MHC-II), the disruption of cell receptors and cluster of differentiation 3 (CD3) signaling, as well as other antigen escape mechanisms [22]. Failure to halt the progression of *M. tb* infection results in its active form. Once active, *M. tb* infection can spread hematogenously, causing miliary TB [26].

### 2.4. M. tb: Risk Factors and Epidemiology

TB risk factors include environmental, socioeconomic, and behavioral elements as well as comorbid conditions which increase the chance of contraction or a latent infection progressing to active disease. People residing in overcrowded living conditions including underserved urban centers and prisoners are at a higher risk of contracting TB. Healthcare workers have an increased chance of encountering actively infected individuals and therefore also make up a high-risk group [27]. Individuals infected with HIV are sixteen times more likely to develop active TB than those who are HIV-negative [7]. Other factors that can impair a person’s immune mediated response also increase the risk of TB, such as a comorbid diagnosis of diabetes mellitus, severe malnutrition, alcohol dependence, or use of a tumor necrosis factor-alpha inhibitor for treatment of inflammatory conditions [27]. In 2021, the WHO estimated that 2.2 million of the new cases of TB were attributable to undernutrition [7]. Furthermore, young children are at a higher risk of contracting severe TB, likely due to differences in their immune systems as compared to those of adults [28].

Overall, people with lower socioeconomic status have a higher chance of exposure to multiple TB risk factors, such as overcrowded living conditions, food insecurity leading to malnutrition, tobacco products, alcohol, and indoor air pollution with poor ventilation. This is especially true in certain regions throughout the Eastern Hemisphere [Table 1]. Furthermore, individuals with lower socioeconomic status frequently encounter barriers to accessing health care, delaying time to diagnosis and therefore treatment of TB. This can lead to more cases of latent TB becoming active, and more opportunities for the infection to spread throughout the community [27].

### 2.5. M. tb: Prevention

The Bacille Calmette–Guerin (BCG) vaccine is the only licensed vaccine which prevents severe forms of *M. tb* infection in children. Currently, there is no licensed TB vaccine for adults [7]. The BCG vaccine has been somewhat controversial, as for many years it was difficult to determine whether the vaccine prevented acquisition of TB or if it instead prevented progression to active infection [29]. Researchers suggested that this difficulty was in part attributed to the fact that the tuberculin skin test used for latent TB testing could not differentiate between a true positive TB infection, a non-tuberculous mycobacterial infection, or previous inoculation with the BCG vaccine. The tuberculin skin test also has further limitations which complicated the discussion in that it has limited sensitivity in immunosuppressed individuals and can lead to false negative results [30]. However, in 2001, an interferon gamma release assay (IGRA) was developed which can detect TB and discriminate infection from inoculation with the BCG vaccine and other mycobacterial infections. With the use of interferon gamma release assays, studies have demonstrated that the BCG vaccine not only prevents acquisition of an *M. tb* infection, but it also prevents progression of the infection to active disease [29]. In 2015, the IGRA was refined and re-introduced as the QuantiFERON-TB Gold Plus test. Although the BCG vaccine is currently only approved for use in children, in 2019, a cross-sectional study by Katelaris et al. suggested that the vaccine is also associated with lower prevalence of latent *M. tb* in adult contacts of TB [31]. This finding may influence future research and direct more expansive immunization programs.

The prevention of TB is additionally supported by systemic screening, preventative treatment, government policies, and continued research [32]. The WHO announced their “End Tuberculosis Strategy” in 2015, an effort that aims to decrease global TB cases through early detection, treatment, and prevention, equal access to affordable care, government accountability, improved vaccination efforts, poverty alleviation, regulatory frameworks for case notification, and promotion of new innovations in TB research [33].

### 2.6. M. tb: Treatment

According to WHO guidelines, first-line treatment for new pulmonary *M. tb* is 2 months of daily isoniazid, rifampicin, ethambutol, and pyrazinamide followed by four months of daily isoniazid and rifampicin. Individuals aged 12 and older with drug-susceptible TB can alternatively be treated with four months of isoniazid, rifapentine, moxifloxacin, and pyrazinamide. Children aged 3 months to 16 years with non-severe pulmonary TB can be treated with a shorter course consisting of 2 months of isoniazid, rifampicin, ethambutol, and pyrazinamide followed by 2 months of isoniazid and rifampin. The WHO also recommends that HIV-positive individuals start antiretroviral therapy within at least 2 weeks of initiating their TB treatment, regardless of CD4 count. In individuals with TB meningitis, adjuvant corticosteroid treatment with a 6–8 week taper of dexamethasone or prednisolone is advised [34]. People with comorbid alcohol dependence, renal failure, diabetes, high risk of developing peripheral neuropathy from malnutrition, pregnant, or who are breastfeeding should also receive vitamin B6 supplementation while taking isoniazid [35].

Research on monoclonal antibodies has demonstrated the potential for TB treatment to reduce treatment time and, therefore, decrease transmission and incidences of antibiotic drug resistance [36]. In 2008, a study by Lopez et al. revealed that in mice infected with *M. tb*, treatment with the monoclonal antibodies TBA61 and TBA84 reduced bacterial load and caused improved histological granuloma organization in the lungs [37]. A different study by Balu et al. in 2011 concluded that intranasal inoculation with human antibody IgA1 constructed with a single-chain variable fragment clone 2E9 was able to reduce TB infection in mice [36]. The 2E9IgA1 monoclonal antibody targets the human CD89 receptor, leading researchers to postulate that other immunomodulators that increase expression of CD89 on target cells could be investigated [36]. Continued research into the potential of monoclonal antibodies as passive immunotherapy for TB may prove to be especially critical for populations that cannot receive active immunization or for those infected with antibiotic-resistant strains of *M. tb* [36].

## 3. Interferons

### 3.1. Type I Interferons: Production and Signaling

Type I IFNs are polypeptide products from a multi-gene cytokine family in innate immune cells that encode 13 partially homologous IFNα subtypes, a single IFNβ product, and poorly defined single IFNε, IFNτ, IFNκ, IFNω, IFNδ, and IFNζ subtypes [38]. This review will mainly focus on IFNα and IFNβ. These type I IFNs are traditionally known for their ability to induce an antiviral state that interferes with at least three stages of the viral replication cycle [39]. IFNα is predominantly produced by hematopoietic cells, such as plasmacytoid dendritic cells, while IFNβ can be produced by most cell types, predominantly fibroblasts and epithelial cells, upon viral infection [40]. IFNs are produced when pattern recognition receptors located on the cell surface, cytosol, or endosomal compartments are stimulated by microbial products such as nucleic acids or non-nucleic-acid pathogen-associated molecular patterns [41].

TLR4 is a cell-surface Toll-like receptor that recognizes lipopolysaccharides from bacteria and leads to a fundamental IRF3 or IRF7 pathway to induce IFNα/β synthesis [Figure 1]. It has been found that transcription of IRF3 induces IFNB and IFNA4 genes, which then triggers IRF7 in a positive feedback loop, with Nuclear factor-κB as a cofactor, for IFNα/β products [42]. In the cytosol, a recently described receptor cytosolic GAMP synthase can identify DNA motifs for IFN production. These cytosolic DNA sensors catalyze cGAMP products in response to DNA, which are recognized by stimulator-of-interferon genes to trigger type I IFN synthesis [43]. 

IFNα/β can bind a heterodimeric transmembrane receptor composed of IFNAR1 and IFNAR2 subunits. IFNAR1 is associated with and activates Janus Kinase 1 (JAK1), while IFNAR2 is associated with and activates Tyrosine Kinase 2 (TYK2). Activated JAK1 and TYK2 phosphorylate cytosolic STAT1 and STAT2, leading to dimerization and nuclear translocation with IRF9, to form the Interferon-stimulated gene factor 3 (ISGF3) complex [44]. This ISGF3 complex can initiate transcription by binding IFN-stimulated response elements in promoter regions. Transcriptional products consist of several hundred ISGs, with varying functions. First, they create an antimicrobial state in infected and non-infected cells to limit spread. Secondly, they boost antigen presentation and NK cell function while balancing to minimize pro-inflammatory pathways. Lastly, they activate T and B cell adaptive immune responses towards developing high-affinity antigen specificity for viral memory [45].

### 3.2. Role of Type II Interferons during M. tb Infection

Type II IFNs are important cytokines of the immune system and cellular immunity that contain one subtype, IFN-γ. IFN-γ has many roles but a key one is activating macrophages to produce a direct antimicrobial and antitumor mechanism. IFN-γ is also involved in leukocyte attraction and functions as an immunomodulator causing immunoglobulin production and class switching [46]. IFN-γ signals primarily via the Jak-STAT pathway after binding to the IFNGR1 and IFNGR2 cell surface receptors that are associated with JAK1 and JAK2, respectively. IFN-γ is secreted by TH1 cells and NK cells in response to microbial infection [44,47,48]. Humans that have loss-of-function genetic mutations in IFN-γ or its receptor have increased susceptibility to mycobacterial infections [49]. CD4+ and CD8+ T cells are primary sources of IFN-γ and are important for host survival in *M. tb* infection [50]. IFN-γ recruits immune cells to the site of infection, participating in granuloma formation, which helps with disease progression control as well as stimulating macrophages and producing the oxidative burst to control growth [51]. Infected macrophages use TLR 1/2 and TLR 2/6 to recognize *M. tb*, which allows macrophages to release IL12 and induce Th1 cells to produce more IFN-γ. IL-12 is induced after phagocytosis of *M. tb* by macrophages and DCs, driving the development of a Th1 response [52]. Type II IFNs are modulators of the Th2 response as continuous production of IFN-γ by Th1 cells inhibits the proliferation of Th2 cells to ensure inflammatory Th1 cytokines are not inhibited [53,54,55]. One of the most significant *M. tb*-controlling effects of Type II IFNs is the production of large amounts of reactive oxygen intermediates (ROI) and reactive nitrogen intermediates (RNI) by macrophages through the actions of an inducible form of nitric oxide synthase (NOS) [56]. IFN-γ synergizes with TNF-α to activate macrophages to stimulate NOS to contribute to the killing of *M. tb* by inducing apoptosis of *M. tb*-infected macrophages via the intrinsic pathway [57]. IFN-γ can also have an anti-inflammatory role of inhibition of neutrophils by diminishing the production of IL-17 [58]. IFN-γ mediates antimicrobial activity through production of lysosomal peptides to facilitate inflammation against microbes [59]. IFN-γ, when in combination with TLR2 stimulation, produces antimicrobial peptides as well as stimulates autophagy in a Vitamin D-dependent pathway [60,61]. Through the Vitamin D3-dependent manner, IFN-γ causes the production of hCAP18/LL-37 cathelicidin to induce autophagy [59].

### 3.3. Other Key Mediators in Host M. tb Immune Response

*M. tb* is an obligate human pathogen that survives and replicates in host macrophages inside a modified phagosomal compartment, thus evading macrophage killing by neutralizing RNIs [62,63]. As previously described, infection with *M. tb* begins with phagocytosis of the pathogen by antigen-presenting cells such as dendritic cells [64]. Initial recognition of *M. tb* is mediated by pattern recognition receptors such as Toll-like receptors that recognize pathogen-associated molecular patterns, which are conserved microbial molecular motifs, thus eliciting both an innate and adaptive immune response by the host [65,66]. *M. tb* can activate immune cells through TLR2 or TLR4 in a CD14-independent manner. Macrophages activate STAT-1 and NF-κB pathways which lead to nitric oxide production, and the TLR-dependent activation of NF-κB is mediated via MyD88, an adaptor protein required for signal transduction through 1L-1R [67,68].

Development of an efficient innate and adaptive immune system is key to controlling an *M. tb* infection and the main mediators are derived CD4+ T cells that produce cytokines IFN-γ, IL-2, and lymphotoxin-α [69]. On the other hand, CD8 T cells act directly on *M. tb* through release of granzymes and perforins that have an effect through Fas ligand-dependent and -independent manners [69,70]. The transfer of the bacterial antigens from the phagosomal compartment into the cytosolic compartment of APCs is sampled by the Major Histocompatibility Complex class I pathway [71]. After presentation, many chemokine molecules are upregulated to drive the migration of infected cells in the lymph nodes [72]. γδT cells play a proinflammatory role in protection against *M. tb* and are a significant source of IL-17 and IFN-γ [58,70]. IL-17 is a cytokine that promotes the inflammation that is caused by neutrophils [58]. Neutrophils have a protective role against *M. tb* early in lung infections by producing IL-1b, TNF-α, defensins, cathelicidins, lipocalin, and NADPH oxidase and superoxides to combat the pathogen [73]. 

Type III IFNs, also known as IFN-κ, found mostly in epithelial cells in the mucosa, have similar biological properties as type I IFNs that include antiviral and antitumor activities. They both become stimulated in a similar manner via PRRs; however, they have a lower kinetic profile than type I IFNs [74,75,76,77,78,79]. *M. tb* infection in lung epithelial cells stimulates upregulation of IFN-κ genes [80]. The signal transduction of type III IFNs is similar to type I IFNs in inducing a JAK/STAT signaling pathway; however, it relies on specific IL-28Ra and IL-10R2 chains [81]. IFN-α and IFN-κ have antiviral and antitumor activities; IFN-γ, on the other hand, has immune mediatory, antibacterial, antifungal, and antiparasitic roles [38]. T cells, macrophages, and DCs secrete TNF-α after *M. tb* induction to control the infection, and, in synergy with IFN-γ, induce NOS2 expression [82,83]. IL-10 is generally an anti-inflammatory cytokine that possesses macrophage-deactivating effects by downregulating IL-12 production and decreasing IFN-g production, which was reversed with IL-10 inhibition [84]. 

Other proposed mediators in *M. tb* infection include chemokines induced by *M. tb* such as RANTES, MIP1-alpha, MIP2, MCP-1, MCP-3, and MCP-5 [85]. These chemokines are important modulators in the formation and maintenance of granulomas in *M. tb* infection. CCR5, the receptor for RANTES, MIP1-alpha, and MIP1-beta, was increased following *M. tb* infection [86]. Glutathione (GSH), an antioxidant, facilitates the control of intracellular *M. tb* growth in both murine and human macrophages with direct antimicrobial activity [87,88]. GSH combined with IL-2 and IL-12 strengthens the NK cell effects to control *M. tb* infections. It has been shown that synthesis of TNF-α and free radicals leads to a decrease in GSH levels in T cells, which causes impaired production of the Th1 cell line of cytokines and enhances the growth of *M. tb* inside macrophages. Treatment of T cells with N-acetylcysteine (NAC), a precursor for GSH, increases the production of IL-12, IL-2, and IFN-γ, which are critical for the control of intracellular pathogens [89,90].

The mTOR pathway is an important regulator of autophagy which has been observed as a mechanism the host uses to combat *M. tb* infection [91,92]. Autophagy induced by rampamycin has shown to increase antigen presentation in DCs and thus increase the efficacy of the BCG vaccine [93]. Autophagy can be activated by Vitamin D3, TLR, or by inhibition of mTOR by rapamycin, and this limits *M. tb* infection [94,95,96,97]. Previous studies of several lipid mediators in *M. tb* infections have shown that leukotriene B4, Lipoxin A4, prostaglandin E2, and prostaglandin F2α play an important role in the pathogenesis of TB. Overall, the leukotriene and prostaglandins were mostly pro-inflammatory and the lipoxins were anti-inflammatory in individuals with TB [98,99].

## 4. Role of Type I Interferons in *M. tb* Infection

### 4.1. Type I IFN Transcriptional Signature in M. tb Infection

Gene signatures in the blood of patients with active TB can be used as a marker for diagnosis, disease manifestation, and treatment monitoring. Berry et al. conducted a landmark study which demonstrated that patients with active TB have a predominantly type I IFN-inducible gene signature that correlates with lung radiographic disease severity and is downregulated following treatment [100]. These findings have since been replicated in numerous studies [101]. Bloom et al. and Wang et al. described that overexpressed IFN-inducible genes tended to be primarily in neutrophils and monocytes, hinting at overactivation and infection of these cell types during *M. tb* infection [102,103]. Studies have demonstrated early overexpression of type I IFN genes in patients with latent TB that eventually progressed to active infection. An enhanced type I IFN signature was correlated with progression to active TB up to 18 months prior to diagnosis [100,104,105,106,107,108]. This suggests that the type I IFN response can precede the onset of active disease and symptoms. Monitoring the type I IFN response to chemotherapy would thus be more advantageous compared to the detection of acid-fast bacilli and provide an earlier assessment in the clinical management of infected patients [109]. Multiple studies also described an increased resistance to TB in individuals with a mutation in IFNAR1 (IFN 1 receptor) that impaired type I IFN signaling [103,110,111]. In addition, patients with a deficiency in the gene encoding ISG15 displayed increased type I IFN responses and were seen to be more susceptible to *M. tb* [112]. Multiple studies have reported reactivation of TB in patients receiving IFN-α-based therapy for chronic viral hepatitis [113,114,115,116,117,118].

### 4.2. Mechanisms of Type I IFN Induction in M. tb Infection: ESX-1 Protein Secretion System and Pattern Recognition Receptors

The outcomes of an infection are determined by the dynamic interplay between virulence factors produced by a pathogen and the immune response mounted by the host. Immune responses are characterized by increased cytokines, chemokines, and other products normally meant to protect the host. Various signaling pathways have been described that induce type I IFN expression in response to *M. tb* infection. The *M. tb* virulence factor ESX-1 protein secretion system is seen to be a major contributor to increased levels of type I IFN. Studies have shown that type I IFN responses were diminished in human macrophages and mice models, although not completely, if the genomic region of difference-1 (RD1) of ESX-1 was deleted [119,120,121,122,123]. ESX-1 disrupts the phagosomal membrane leading to mitochondrial stress and resultant leakage of mitochondrial DNA and mycobacterial products (such as early secreted antigenic target-6 and culture-filtrate protein-10) into the cytosol [121]. These products are recognized by cyclic GMP-AMP synthase (cGAS), which then initiates the formation of a second messenger, cyclic-GMP-AMP (cGAMP) [Figure 2]. cGAMP and bacterial cyclic dinucleotides interact with STING or STING-accessory molecule DDX41. Activation of STING results in its relocation to the perinuclear Golgi, where it then initiates recruitment and activation of TANK-binding Kinase 1 (TBK1). TBK1 activates NF-kB, which mediates the induction of pro-inflammatory cytokines, as well as phosphorylates interferon regulatory factors IRF3 and IRF5, both of which facilitate type I IFN expression [108,120]. A study conducted by Wiens and Ernst demonstrated that the extent of mitochondrial DNA released during *M. tb* infection may determine the extent of type I IFN expression [124]. cGAMP can also access uninfected cells via gap junctions leading to STING activation and type I IFN expression in nearby bystander cells [123,125]. Both cGAS- and STING-deficient human and mouse macrophages are impaired in their ability to express IFN-β in response to *M. tb* infection [123,126,127,128]. Toll-like receptors (TLR) are transmembrane proteins that serve as pattern recognition receptors and can sense *M. tb* components [129]. The ligands of different *M. tb* strains were seen to dimerize different TLRs, resulting in intracellular signaling and varying macrophage responses that drive the production of type I IFN [130,131]. Other pattern recognition receptors including Nod-like receptor (NOD2) have also been linked to increased type I IFN expression [120,132]. NOD2 recognizes the mycobacterial product N-glycosylated muramyl dipeptide, which leads to RIP2-mediated activation of TBK1, resulting in downstream type I IFN expression via IRF5 dimerization and nuclear translocation. In a study performed by Pandey et al., this pathway was significantly diminished in infection with ESX-1-deficient *M. tb* [120]. In summation, extensive literature reveals that different *M. tb* strains induce varying type I IFN responses via multiple mechanisms and signaling pathways which may result in differing virulence.

### 4.3. Mechanisms of Type I IFN Induction in M. tb Infection: ESX-1 Protein Secretion System Independent Mechanisms

As mentioned earlier, type I IFN expression is reduced but not eliminated in ESX-1-deficient *M. tb*. The attenuated *Mycobacterium bovis* strain Bacillus Calmette–Guérin (BCG) lacks the RD-1 region that encodes the ESX-1 secretion system, yet this strain still produces a type I IFN response in human macrophages [119,126]. This response was reduced in STING-cells, hinting that this is a STING-dependent process and that mycobacteria components can still gain access to the cytosol in the absence of ESX-1.

### 4.4. Pathogenic Effects of Type I IFN Signaling during M. tb Infection

How type I IFNs exacerbate *M. tb* infection is not yet fully understood, but many mechanisms have been described in various studies. Type I IFN inhibition of protective cytokines (IFN-γ, TFN-α, IL-12) and Th1 cell responsiveness in human cells was described in multiple studies and thought to be mediated by impairment of IL-12 production [119,133,134,135]. Mouse models infected with the hypervirulent HN878 *M. tb* strain were particularly seen to have reduced TH1 responses and lower IFN-y production secondary to a high type I IFN response [136,137,138]. Multiple studies have also shown that type I IFNs limited human and mouse mycobacteria-restricting monocyte/macrophage response to the antibacterial effects of IFN-γ during *M. tb* infections and this process was mediated by IL-10, a cytokine that can impair antimycobacterial immune responses during infection [133,134,139]. IL-10 inhibits the IFN-γ-driven vitamin D3/cathelicidin/defensin beta 4A antimicrobial defense pathway [134]. In addition, type I IFNs have also been shown to promote early cell death of alveolar macrophages and increase accumulation of myeloid cells which contribute to the spread of infection and lung inflammation [110,140]. Data from the study by Moreira-Teixeira et al. demonstrated that increased type I IFN signaling promoted pathogenesis of *M. tb* infection in mouse models via stimulation of neutrophils and a pathological neutrophil extracellular trap (NET) response [141]. NETs are networks of extracellular fibers composed of DNA from neutrophils which bind pathogens and are typically involved in an antimicrobial response. In *M. tb* infection, NETosis (NET activation and release) contributed to disease exacerbation in this study. Type I IFNs have been reported to inhibit the production of IL-1α and IL-1β, which are important for host defense against *M. tb* infection in humans and mice [119,133,135,142,143,144,145]. This inhibition was shown to be dependent on NO synthase 2 and IL-10 [119,135,139]. NO interference with the NLRP3 inflammasome assembly impairs IL-1β processing by *M. tb*-infected macrophages [146]. Prostaglandin E2 (PGE2) serves as a mediator of IL-1-dependent host protection and is known to prevent necrosis of *M. tb*-infected macrophages by promoting apoptosis which limits pathogenic dissemination [147,148]. Type I IFNs are shown to inhibit PGE2, and this could thus promote necrosis and alter the ratio of host-protective PGE2 to host-detrimental 5-lipoxygenase products such as lipoxin A4 [149,150]. Animal model studies revealed evidence that suggests the degree of *M. tb*-induced type I IFN expression in mice is correlated with the virulence of *M. tb* strains [124,136]. In a study performed by Manca et al., administration of anti-IFNα/β antibodies to mice before and upon infection with *M. tb* was associated with a long-term survival benefit but no significant change in bacterial burden [137]. This was further affirmed in studies which demonstrated that administration of IFN-α/β in mice impaired host survival and worsened lung inflammation in the face of *M. tb* infection [137,140,149]. Dynamic cytokine interactions mediate the pathogenic and protective functions of various molecules in *M. tb* infection, and a greater understanding of these key mediators opens the door for host-directed therapeutic intervention in *M. tb* infection. 

### 4.5. Potential Protective Functions of Type I IFN in M. tb Infection

There is emerging evidence that type I IFNs can display protective functions in *M. tb* infection under certain conditions. Ward et al. and Bax et al. demonstrated that patients with IFN-γ receptor signaling deficiencies who failed conventional TB antimycobacterial chemotherapy (consisting of isoniazid, rifampicin, ethambutol, and pyrazinamide) or had recurrent disease benefited from inhaled or subcutaneous coadministration of IFN-α in addition to standard TB antimycobacterial chemotherapy [151,152]. Further studies described that in the absence of IFN-γ signaling, type I IFNs may contribute to the survival of myeloid cells that help control *M. tb* pathogenesis [130,153]. These findings may explain some of the mechanisms underlying the beneficial effects of IFN-α treatment in patients with compromised IFN-γ responses. Rivas-Santiago and Guerrero showed that administration of IFN-α during BCG vaccination promoted production of protective cytokines (IFN-γ, TNF-α, IL-12) and provided increased protection against *M. tb* infection than that with BCG alone [154]. McNab et al. further demonstrated that basal type I IFN signaling is necessary for the maximum production of IL-12 and TNF-α by macrophages in response to *M. tb* [139]. The findings presented suggest that an understanding of balance between type I and type II IFNs is necessary to define the ability of the host to control *M. tb* infection. The interplay of these molecules could be harnessed to develop interventions against infection.

## 5. Conclusions

While the current understanding of the effects of type I IFN signaling on *M. tb* infection remains unclear, further research is being performed in both mouse and human models that could have strong therapeutic implications in the future. Type I IFNs have well-defined roles in modulating host immune response against viruses and their potential pathogenicity in *M. tb* infections has been seen in other bacteria species as well. Multiple studies have demonstrated that *Mycobacterium leprae*, *Mycobacterium bovis*, and *Listeria monocytogenes* induce type I IFN production by infected macrophages contributing to their pathogenesis [103,124,155]. Type I IFN signaling is one of many key mediators that have been identified in the host response to *M. tb* infection and warrant further exploration as potential targets for host-directed therapy in adjunct to antibiotics for resistant *M. tb* infection. Overall, our findings relay that while some instances have been seen of type I IFN expression contributing to host defense against *M. tb* infection, most studies relate type I IFN signaling to pathologic responses in TB infection. Nevertheless, increased understanding of type I IFN-driven processes could lead to the development of further interventions against this major global health problem.

## Figures and Tables

**Figure 1 ijms-24-03919-f001:**
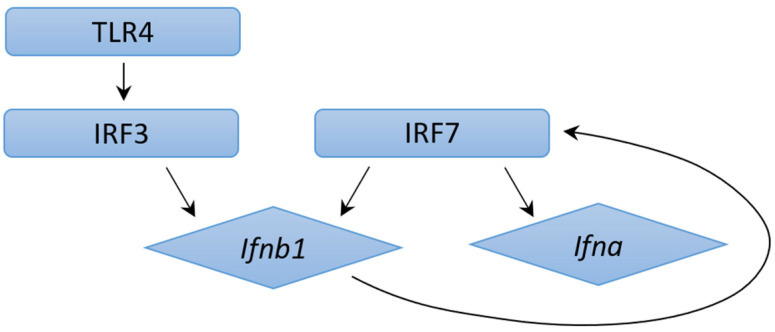
IFNs expression through IRF3 and IRF7 pathways.

**Figure 2 ijms-24-03919-f002:**
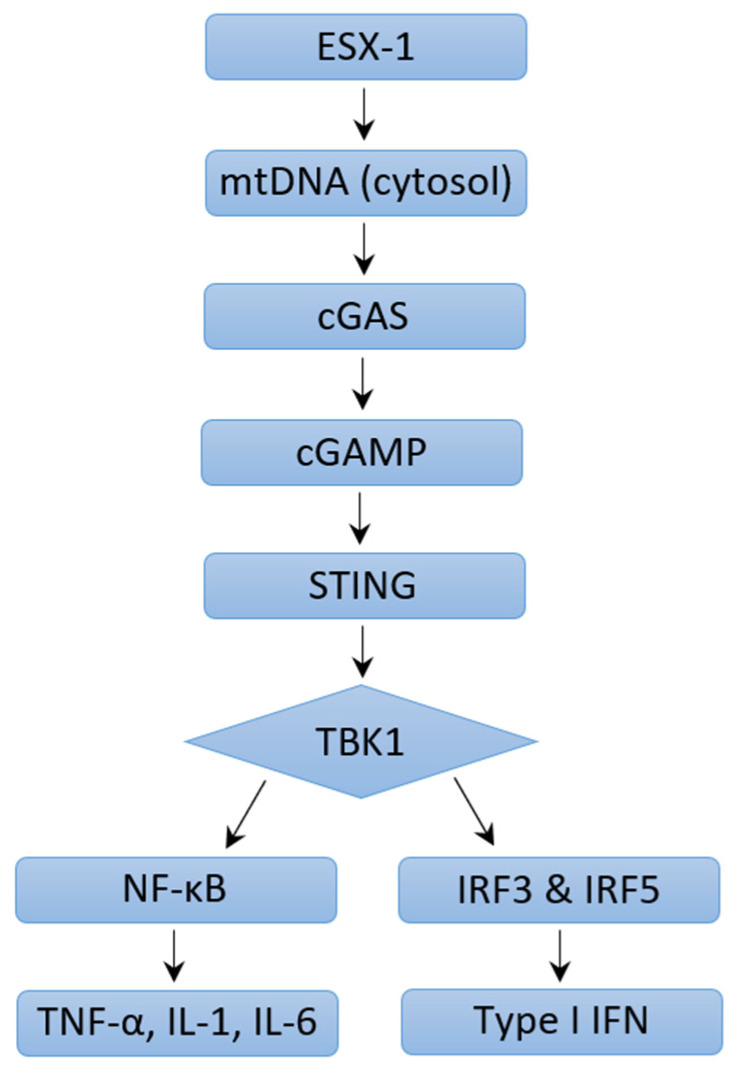
Mechanisms of Type I IFN Induction through ESX-1 pathway.

**Table 1 ijms-24-03919-t001:** Epidemiology of tuberculosis in 2021 [7].

**Age/Sex Group**	**Number of Cases**	**Percent Total**
Adult men	6,000,000	56.5%
Adult women	3,400,000	32.5%
Children	1,200,000	11%
**Region**	**Number of Incidences**	**Percent Total**
Southeast Asia	630,000	45%
Africa	322,000	23%
Western Pacific	252,000	18%

## Data Availability

Not applicable.

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
