# Peer review of "Pathogenicity of Type I Interferons in Mycobacterium tuberculosis"

_ijms, 2023, doi:10.3390/ijms24043919_

Round 1
Reviewer 1 Report
While Type II IFN (IFN-γ) is known to play a significant role in the course of tuberculosis (TB), the importance of Type I IFNs in mycobacterial infection is much less recognized. From this point of view, the evaluated article is a valuable contribution to systematizing recent reports relating to the role of Type I interferons in TB. However, the manuscript needs intensive rewriting/rearranging. In general, the text has been divided into too many parts. This applies in particular to Chapter 2, which contains as many as 16 subsections.
Major remarks:
1/ please consider whether subsection 2.5. is absolutely necessary, I suggest in a much shortened form to include it in the Introduction, without carving out a separate section
2/ rename Chapter 2 "Results", why this name?
3/ combine 2.1 with 2.2 into one subsection and limit the content to only the most important information; include information on what the percentage of patients with the pulmonary form of tuberculosis and various extrapulmonary forms is now worldwide (or at least in the US, since the authors have US university affiliations)
4/ line 97: did the authors really mean that IFN-γ kills macrophages? The term is inaccurate, to say the least. Interplay between IFN-γ and macrophages in inflammation and acquired immunity during infection is crucial to the host’s ability to mount an effective immune response. It has been discussed in detail by Schroder et al. (2013, J. Leuc. Biol. https://doi.org/10.1189/jlb.0603252).
5/ lines 384-386 contain the purpose of the paper; as a rule, the purpose of the paper should be in the Introduction, but more importantly it should not contain wording that directs the reader's train of thought toward a preconceived thesis (here that "...Type I IFNs play a pathogenic role in M. tb infection"), especially since the very next sentence contradicts this formulation, and the information in subsection 2.16 and in the Conclusions (lines 516-517) stands in opposition to this thesis; the thesis/goal should be rephrase.
6/ line 460: the term "protective cytokines" is imprecise; do they mean the same cytokines cited in line 508?
7/ line 501: conventional TB therapy includes the use of chemotherapeutics and so it is still "antimycobacterial chemotherapy”. Does chemotherapy used in combination with type I IFNs consist of a combination of other antibiotics than that described in lines 225-229?
Minor remarks:
1/ line 27: the abbreviation “M.tb” is written in italics, while in the rest of the paper it is written in regular font, please unify the notation
2/ line 35: are there more recent data regarding the number of LTBI cases worldwide than those from the 2014 WHO report? Cohen et al. (2019, Eur. Respir. J.) in their meta-analysis estimated that one-fourth of the world's population is latently infected with TB (DOI: 10.1183/13993003.00655-2019).
3/ lines 57-59 ("...there is still much to be discovered in the understanding..."), 118-120 ("Much research still needs to be done..."), 191-192 ("This finding may influence future research..."): overly vague statements are repeated too often
4/ lines 61-62 ("...it is not exclusively a lung disease..."), line 90 ("Dendritic cells...travel to..."), line 107 (“… can kick in, resultingin…”): replace these phrases/expressions with more formal language
5/ line 294: what does the abbreviation "IG" mean? I'm guessing it's about immunoglobulins, but the standard abbreviation is "Ig"
6/ line 310: put the abbreviation "ROI" here and expand the abbreviation “RNI”; replace with this abbreviation its deciphering in line 323
7/ line 312: the notation "TNF-alpha" appears again, while already in line 87 it was replaced by "TNF-α”
8/ Line 430: is - "A study conducted by Wiens demonstrated...", should be - "by Wiens and Ernst".
Author Response
Dear Reviewers,
We appreciate the time and effort you have all taken to provide an extensive and thorough review of our manuscript and agree that the suggested edits have greatly improved it. The objective of our paper was to gather evidence from human case reports and animal models to explore the pathogenic and potentially protective roles that Type 1 IFNs play in M. tb infection. We hope that our revised manuscript based on all the suggested edits make this objective much clearer.
Thank you all for you valuable time.
REVIEWER#1
While Type II IFN (IFN-γ) is known to play a significant role in the course of tuberculosis (TB), the importance of Type I IFNs in mycobacterial infection is much less recognized. From this point of view, the evaluated article is a valuable contribution to systematizing recent reports relating to the role of Type I interferons in TB. However, the manuscript needs intensive rewriting/rearranging. In general, the text has been divided into too many parts. This applies in particular to Chapter 2, which contains as many as 16 subsections.
Major remarks:
1/ please consider whether subsection 2.5. is absolutely necessary, I suggest in a much shortened form to include it in the Introduction, without carving out a separate section
Thank you for your feedback, we agree with this assessment, and have deleted subsection 2.5 entirely. Instead, we have briefly included some epidemiology in the introduction, and include a chart further breaking down some of the information that was formerly in the epidemiology section.
2/ rename Chapter 2 "Results", why this name?
We agree with this comment and have broken down the paper into further chapters. All content containing background information on M. Tb has now been denoted under Chapter 2 (Background). Sections pertaining to background information on Interferons have been denoted under Chapter 3 (Interferons), and sections specifically addressing type 1 interferons pathogenicity have been placed under Chapter 4 (Role of Type 1 Interferons in M. Tb infection).
3/ combine 2.1 with 2.2 into one subsection and limit the content to only the most important information; include information on what the percentage of patients with the pulmonary form of tuberculosis and various extrapulmonary forms is now worldwide (or at least in the US, since the authors have US university affiliations)
Thank you for this feedback. This section was combined and cut down on excessive information. Additionally, regarding epidemiology, we discuss this in more detail in the section on epidemiological information. However, we added a brief statistic here discussing percentages of pulmonary vs extrapulmonary patients from the CDC website which is a public domain.
4/ line 97: did the authors really mean that IFN-γ kills macrophages? The term is inaccurate, to say the least. Interplay between IFN-γ and macrophages in inflammation and acquired immunity during infection is crucial to the host’s ability to mount an effective immune response. It has been discussed in detail by Schroder et al. (2013, J. Leuc. Biol. https://doi.org/10.1189/jlb.0603252).
Thank you for catching this misinformation, we agree with your statement that IFN-y is not responsible for killing macrophages. Rather CD4+ T-cells, which produce IFN-y, act independently of IFN-y to control the growth of M. Tb infection via nitric oxide-dependent mechanisms. (S. Ahmad, “Pathogenesis, immunology, and diagnosis of latent mycobacterium tuberculosis infection,” Clinical and Developmental Immunology, vol. 2011. 2011. doi: 10.1155/2011/814943.) We made changes found on lines 97-100. Further, we added that IFN-y helps augment the immune response to infection.
5/ lines 384-386 contain the purpose of the paper; as a rule, the purpose of the paper should be in the Introduction, but more importantly it should not contain wording that directs the reader's train of thought toward a preconceived thesis (here that "...Type I IFNs play a pathogenic role in M. tb infection"), especially since the very next sentence contradicts this formulation, and the information in subsection 2.16 and in the Conclusions (lines 516-517) stands in opposition to this thesis; the thesis/goal should be rephrase.
Thank you for this suggestion! We agree with and understand what you are saying and deleted the first four sentences of section 2.12 to eliminate any confusions/contradictions.
6/ line 460: the term "protective cytokines" is imprecise; do they mean the same cytokines cited in line 508?
Thank you for catching this. We have placed “IFN-γ, TNF-α, IL-12” in parenthesis after the term protective cytokines to clear this up.
7/ line 501: conventional TB therapy includes the use of chemotherapeutics and so it is still "antimycobacterial chemotherapy”. Does chemotherapy used in combination with type I IFNs consist of a combination of other antibiotics than that described in lines 225-229?
Thank you for this suggestion. We have reworded this section to clarify that IFN- α in conjunction with standard TB antimycobacterial chemotherapy (consisting of isoniazid, rifampicin, ethambutol, and pyrazinamide) yielded benefits according to the article mentioned.
Minor remarks:
1/ line 27: the abbreviation “M.tb” is written in italics, while in the rest of the paper it is written in regular font, please unify the notation
Thank you for catching this, we have corrected this mistake.
2/ line 35: are there more recent data regarding the number of LTBI cases worldwide than those from the 2014 WHO report? Cohen et al. (2019, Eur. Respir. J.) in their meta-analysis estimated that one-fourth of the world's population is latently infected with TB (DOI: 10.1183/13993003.00655-2019).
3/ lines 57-59 ("...there is still much to be discovered in the understanding..."), 118-120 ("Much research still needs to be done..."), 191-192 ("This finding may influence future research..."): overly vague statements are repeated too often
Thank you for taking the time to revise our paper. We went ahead and deleted lines 118-120 to prevent repeated statements.
4/ lines 61-62 ("...it is not exclusively a lung disease..."), line 90 ("Dendritic cells...travel to..."), line 107 (“… can kick in, resulting in…”): replace these phrases/expressions with more formal language
We appreciate your response and agree that we need to use more formal language. We went ahead and changed “travel to” to “migrate towards” on line 90 and removed “can kick in” on line 108-109 to make the sentence more formal. We also changed the verbiage in lines 61-62 from “exclusively” to “solely” to make it sound more formal.
5/ line 294: what does the abbreviation "IG" mean? I'm guessing it's about immunoglobulins, but the standard abbreviation is "Ig"
Thank you for catching this! We have replaced “IG” with “immunoglobulins”.
6/ line 310: put the abbreviation "ROI" here and expand the abbreviation “RNI”; replace with this abbreviation its deciphering in line 323
Thank you for this suggestion. We have edited this as you recommended.
7/ line 312: the notation "TNF-alpha" appears again, while already in line 87 it was replaced by "TNF-α”
Appreciate you catching this. We have replaced TNF-alpha with TNF-α.
8/ Line 430: is - "A study conducted by Wiens demonstrated...", should be - "by Wiens and Ernst"
Thank you for correcting this. We have edited it as you suggested.

Reviewer 2 Report
The authors based their study on the premise that many factors determining the severity and progression of tuberculosis (TB) remain unclear, even though it is a major cause of mortality due to infectious disease. On the other hand, they emphasize that type I interferons have diverse effector functions regulating innate and adaptive immunity during infections. The present review examines a wide range of studies and concludes that a delicate balance of immunomodulators dictates the host response to TB and that further studies are needed to understand how type I interferons play a role in this response.
The paper has a sound scientific basis and clear problem identification. Nevertheless, how the authors address the basic question of their review (included in the title) is confusing. For instance, the information in subsections 2.1 to 28 (pages 2-6) is excessive and redundant, which can be summarized and consolidated with the support of comparative tables and other resources that facilitate reading. The same applies to the rest of the manuscript, where the question should be addressed properly. Therefore, it is advisable to refocus the paper on the original idea and reformulate it, accordingly, highlighting the gained knowledge and its scientific impact.
Given the above, the current version of the paper is not endorsed for publication in IJMS. However, authors are invited to amend it and resubmit the improved version.
Author Response
Dear Reviewers,
We appreciate the time and effort you have all taken to provide an extensive and thorough review of our manuscript and agree that the suggested edits have greatly improved it. The objective of our paper was to gather evidence from human case reports and animal models to explore the pathogenic and potentially protective roles that Type 1 IFNs play in M. tb infection. We hope that our revised manuscript based on all the suggested edits make this objective much clearer.
Thank you all for you valuable time.
REVIEWER#2
The authors based their study on the premise that many factors determining the severity and progression of tuberculosis (TB) remain unclear, even though it is a major cause of mortality due to infectious disease. On the other hand, they emphasize that type I interferons have diverse effector functions regulating innate and adaptive immunity during infections. The present review examines a wide range of studies and concludes that a delicate balance of immunomodulators dictates the host response to TB and that further studies are needed to understand how type I interferons play a role in this response.
The paper has a sound scientific basis and clear problem identification. Nevertheless, how the authors address the basic question of their review (included in the title) is confusing. For instance, the information in subsections 2.1 to 28 (pages 2-6) is excessive and redundant, which can be summarized and consolidated with the support of comparative tables and other resources that facilitate reading. The same applies to the rest of the manuscript, where the question should be addressed properly. Therefore, it is advisable to refocus the paper on the original idea and reformulate it, accordingly, highlighting the gained knowledge and its scientific impact.
We appreciate your thoughtful evaluation and understand that these subsections were not contributing to the main thesis of the manuscript. These subsections have been extensively condensed and only pertinent information remains. We also went ahead and changed the title of the review to be clearer and more concise regarding the content contained in the paper.

Round 2
Reviewer 2 Report
The authors appropriately addressed the recommendations made during the review of the first version of the manuscript, noting an improvement in the resubmitted version. Hence, I do endorse this paper for publication in IJMS.